# Influences of Growth Species and Inclusions on the Current–Voltage Behavior of Plasma Electrolytic Oxidation: A Review

**Dah-Shyang Tsai** [1,*] and **Chen-Chia Chou** [2]

[1] Department of Chemical Engineering, National Taiwan University of Science and Technology, 43 Keelung Road, Section 4, Taipei 10607, Taiwan

[2] Department of Mechanical Engineering, National Taiwan University of Science and Technology, 43 Keelung Road, Section 4, Taipei 10607, Taiwan; ccchou@mail.ntust.edu.tw

* Correspondence: dstsai@mail.ntust.edu.tw

**Abstract:** Plasma electrolytic oxidation (PEO) has attracted increasing attention since the transportation industry adopts more lightweight metal components and requires an improved version of anodizing for surface protection. In response to the demand, researchers enrich the technical connotation of PEO through diversifying the growth paths and adopting new precursors. Foreign electrolyte additives, involving ceramic and polymeric particles, organic dye emulsions, are incorporated to accomplish various goals. On the other hand, significant progress has been made on comprehension of softening sparks; denoting the adverse trend of growing discharge intensity can be re-routed by involving cathodic current. *I–V* response shows the cathodic pulse current not only cools down the ensuing anodic pulse, but also twists the coating conductivity, and the residuals of twists accumulate over a long time frame, plausibly through oxide protonation. Thus, the cathodic current provides a tool to control the discharge intensity via integration of the coating conductivity deviations. So far, these cathodic current studies have been performed in the electrolytes of KOH and $Na_2SiO_3$. When exotic additives are included, for example $Cr_2O_3$, the cathodic current effect is also shifted, as manifested in remarkable changes in its current–voltage (*I–V*) behavior. We anticipate the future study on cathodic current influences of inclusion shall lead to a precise control of micro arc.

**Keywords:** coating; plasma electrolytic oxidation; inclusion; ceramic; polymer; dye; cathodic current; discharge; electrochemical rectification

## 1. Introduction

Plasma electrolytic oxidation (PEO), widely known as micro arc oxidation (MAO) among Asian countries, may be viewed as an upgraded version of anodizing that produces a sturdy coating to protect the metals. Unlike anodizing, the growth process of PEO is featured with numerous microdischarges traveling on the metallic surface. These breakdown events collectively establish a surface plasma and produce light emissions and acoustic vibrations. Electric discharges, water electrolysis, electrochemical rectification, and electrochemical reactions all take part in building up an oxide coating of high adhesion strength and porosity. Meanwhile, these dielectric breakdowns make the current–voltage (*I–V*) of the electrolytic cell fluctuate, and the coating microstructure difficult to control. In the wake of these microdischarges, dispersed particles of either organic or inorganic origin can be incorporated, providing a separate growth pathway. Inclusions may be used to achieve various purposes, especially those not associated with oxides. The purposes involve the surface aesthetics, surface friction reduction, medical material requirements, additional corrosion resistance, and wear resistance. In the vast literature of PEO, several reviews are worthy of particular mention. The nature of electric discharges and their microstructural effects are reviewed by Clyne [1]; coating process diagnosis and future control

measures are delineated by Yerokhin and Matthews [2]; soft sparking phenomena and their related issues are discussed by Tsai and Chou [3]; processing features and industrial applications are summarized by Simchen and coauthors [4]; growth, structure features, wear and corrosion properties are extensively reviewed by Kaseem and coauthors [5]. In addition, particle inclusions of Al-alloy PEO are reviewed by Lu and Blawert [6], also Borisov [7], while the inclusions of Mg-alloy PEO are summarized by Fattah-Alhosseini [8]. One rarely-mentioned application is reviewed on particle inclusions for the antibacterial PEO surface [9]. Our discussion shall focus on various aspects of growth path and growth species, diagnosis of the micro arc state, the valve effects of including dye colloids and polymer particles, and their connections with plasma control.

Under proper conditions, PEO is capable of producing a thick coating of porous oxide to protect the metal, using a relatively simple apparatus. The coating thickness can reach 200–300 μm, and its porosity allows a variety of modifications. Despite its downside of high electricity consumption, PEO has progressively been accepted as a surface treatment of lightweight metals, especially the widely-used Al-, Mg-, and Ti-alloys. If the workpiece is not made of these three metals, its surface can be converted into one with a metal overlay. For example, on top of the steel workpiece, dip coating aluminum metal makes the steel surface ready for PEO. The as-grown PEO oxides are adequate for the first coat in the sectors of aviation, defense, and ship industries. These strong and porous coatings can be applied to protect dental and bone implants as well.

In principle, PEO is applicable for all valve metals. Yet the surface treatment is most successful on Al- and Mg-alloys, not so much on Ti-alloys and other valve metals, especially in the sense of industrial development. PEO adequacy goes beyond electrochemical rectification (valve effect) of an oxide/metal interface, also involves a thermodynamic competition edge for oxygen [1] and the fine quality of native oxide. A high enthalpy change of metal oxidation ensures the metal oxide is stable in a hydrogen abundant environment. Since oxygen prefers metal bonding to hydrogen bonding, the metal is oxidized first during water electrolysis, and stable in its oxide form even under negative polarization. The valve effect further accentuates the drive toward metal oxidation. The valve effect here refers to asymmetric $I$–$V$ attributes of the oxide/metal interface [10,11]. The interface of anodized aluminum is an excellent example, displaying a significant resistance when polarized positively, and a low resistance when polarized negatively. Under constant current settings, positive polarization gives rise to a high voltage that creates a strong impetus toward oxidation. On the other hand, the voltage is low under negative polarization, a weak reduction impetus. The aqueous electrolytes are usually alkaline, containing sodium or potassium salts. The presence of $Na^+$ and $K^+$ promotes electrochemical rectification of the oxide coating, and assists in crystallization of the coating oxide due to high positive voltage [12]. Other monovalent cations, or divalent cations diminish the valve effect and increase resistance on negative polarization. Cutback of the valve effect raises the probability of cathodic discharges which are generally detrimental to the coating quality, also deters crystallization of the coating oxide [13,14]. The occurrence of cathodic discharges implies the coating is under highly reducing conditions. Sometimes, the coating may require mild reduction to alter its micro arc state, yet reaching the level of cathodic discharges goes beyond righting a wrong.

The common pH value of electrolytes is around 11–12, not because the surface treatment cannot proceed in neutral or acidic solutions. The selection of alkaline electrolytes is due to a few benefits; involving abundant hydroxyl anion ($OH^-$) for metal oxidation and less eco-impacts when disposing the alkaline solutions, in comparison with acidic solutions. If necessary, neutral or acidic solutions can be employed as electrolytes. For example, researchers often choose acidic or neutral electrolytes containing fluoride anions in Mg-alloy PEO to deposit a layer of mixed oxides and fluorides, which enhance corrosion resistance of the coating [15].

## 2. Imposing Current or Voltage Restriction?

### 2.1. Constant Current Mode

PEO of Al- and Mg-alloys is usually performed galvanostatically, since the constant current setting allows a long sparking period of relatively steady voltage for investigation and control. After all, electric discharges are chaotic events, often in cascades, nearly unpredictable. Discharges in an aqueous solution require high voltage to overcome the dielectric strengths of gas envelope and metal oxide. Typical voltage exceeds 400 V for Al-alloys, 200 V for Mg-alloys, and 250 V for Ti-alloys. A combination of high voltage and high current leads to high electricity consumption. Therefore, the rated power of electricity supply dictates the maximum PEO current permitted. Since the current density is also confined, the minimum current density restricts the area of metal surface to be treated. The typical current density ranges from 30 to 100 mA·cm$^{-2}$ for Al-alloys. The current density can be lower on Mg-alloys, 20–50 mA·cm$^{-2}$. The low limit of current density is due to the prerequisite of sustaining an overall plasma. The high limit is necessary to prevent the breakdown events from growing too violent, too rapidly. To circumvent the high electricity burden, an option is to perform oxidation in the electrolyte of molten salts, which permits microdischarges at low as 30–50 V [16,17]. Such a low operation voltage suggests a great saving on electricity. Nonetheless, our knowledge on PEO in molten salts is incomplete at this stage, hence this discussion shall be directed to aqueous electrolytes only.

Figure 1 shows two typical voltage–time curves of Mg-alloy PEO with a pulsed current of bipolar waveform [18]. Stage (I) is defined as the period in which the metal piece glows without discharges. The voltage arises steeply as the surface oxide grows. Stage (II) is marked with the appearance of oxygen bubbles, along with a pause in voltage increase. The voltage pauses because water electrolysis emerges and creates an extra path for electricity. The stages (I) and (II) are anodizing. It is essential to have a fully covered barrier layer during anodizing. Otherwise, a portion of electricity shall be directed to water electrolysis later, instead of oxide growth with rectification. Stage (III) can be identified with a second rise of less steep slope because microdischarges appear. The discharge events are associated with oxide crystallization and pore development. The stages (II) and (III) may be considered as just one stage, instead of two, since the stage (II) is very brief. Stage (IV) is characterized with many bubbles and a much decelerated voltage increase, accompanied with intense light and acoustic emissions due to breakdown events [19–22].

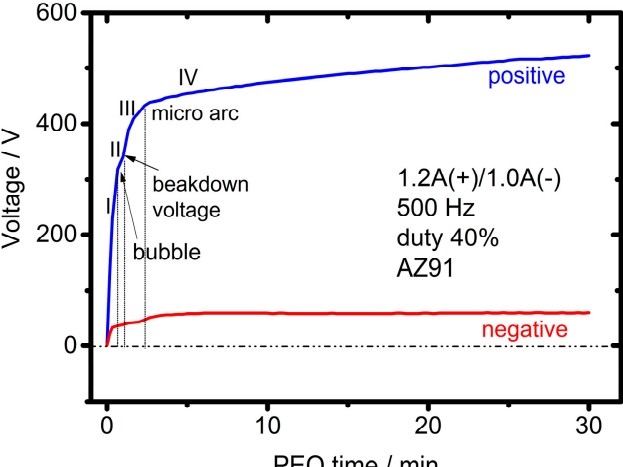

**Figure 1.** Typical voltage–time curves of PEO with a pulsed bipolar current at 500 Hz. Reprinted with permission from [18]; Copyright 2015 Elsevier. The experiment has been performed on an AZ91 plate sample of surface area 27.5 cm$^2$ with 5 g·dm$^{-3}$ Na$_2$SiO$_3$, 1 g·dm$^{-3}$ Na$_2$B$_4$O$_7$, 1.5 g·dm$^{-3}$ KOH in electrolytic solution. Data are plotted as the absolute values of voltage.

Duration of the stage (IV) is generally much longer than its preceding stages, allowing detailed observation and analysis. Most of PEO understandings have been concluded, based on the experimental data of stage (IV), since the voltage is nearly stable and the current is held constant. PEO operations ought to be terminated at a later time of (IV). Otherwise, electrochemical oxidation may proceed to the stage (V), which should be avoided. As the voltage and the coating thickness increase in stage (IV), more discharge occurrences move from the surface into the pores. More discharges in pore give rise to a high probability of coating damages, which ultimately leads voltage decline, that is, the stage (V). In brief, the electric discharge of PEO is a two-edged sword. Electric discharges accelerate oxide growth and consolidate its microstructure at the early stage of growth, at the meantime, the risk of discharge damaging also increases. Therefore, researchers intend to re-orient the hostile trend of discharges to prolong the stage (IV).

The stage (IV) is a period featured with a large number of discharge events on the metal surface under the background of a slowly rising voltage. The common trend in stage (IV) is that the areal number density of sparks decreases with time, while a fraction of sparks increase significantly in size and the rest either increases a little or remains the same size. The size increasing trend is more evident when the duty ratio is set at a high value (80%); less obvious, when the duty ratio is low (10%–20%) [23]. Growth of spark size is evident to the naked eyes, and a large spark is generally viewed as a discharge event of high intensity. Another crucial factor is the discharge location is moving into the pores. When the discharge location is within a pore, the event threatens the coating integrity due to the generated pressure. Therefore, the probability of discharge damage definitely increases with PEO time. To prevent the danger, at certain point in stage (IV), "softening" the discharges is desired; that is, engineering a voltage drop [24–26]. In fact, when positive current $J_+$ is set lower than negative (cathodic) current $J_-$, $0.8 < J_+/J_- < 1$, researchers can intentionally create a "soft sparking" transition in stage (IV). Hence the adverse trend of spark intensity increase is re-routed without changing the electrical parameters manually. The voltage drop subdues the spark intensity, hence, the treatment may extend over a longer period without harming the coating microstructure. At the end of treatment, a thick oxide coating is obtained with a dense inner layer [24,27].

### 2.2. Constant Voltage Mode and Devised Waveform

Compared with the constant current mode, the constant voltage mode is less adopted by researchers. Since the imposed voltage is fixed and the coating thickness grows rapidly in the beginning, the operating current decreases. Normally, the electric current, with a brief rising period, drops one order of magnitude in a nearly exponential manner from its maximum value [28,29]. Thus, the metal surface is subject to a short duration of high anodic current. Besides, in the beginning, the spark number is many, and the spark size is tiny, the probability of discharge damages is relatively low. As the coating thickness increases, electric discharges have less opportunity to grow in intensity since the current is decreasing. Hence an evident benefit of constant voltage mode is to avoid discharge damages or oxidation of non-oxide inclusion. The shortcoming of constant voltage mode is its limited coating thickness.

A combination of constant current and constant voltage modes may help. The designed waveform of *I–V* can be regulated with precision using modern circuit design. Rogov [30] has shown the mode of controlled current with restricted voltage may be useful for PEO treatment of workpieces of complex geometry, since they might have corona discharges at specific locations. Or a power supply, with a smart waveform generator, can generate the pulsed current trains with pulse width control capability. The key problem is on our insufficient knowledge concerning the complex connections between imposed electrical parameters, coating microstructure, and quality.

### 3. Pulsed Current and Cathodic Current Effects

Why do we design a pulsed waveform with periodic recesses and a cathodic current component? When the sample surface is treated with a nonstop DC current, the operation rapidly enters a self-destructive state of stage (V) due to uprising spark intensity. When the anodic DC current is sent with periodic recesses, the unipolar mode, the rising of spark intensity is regularly interrupted. It takes a longer period to reach the state of destruction since the discharge cascade shall be suspended in the recess period, and the trend of growing spark is also redirected. When the current is sent as AC or DC pulsed current of bipolar waveform, the cathodic current shall reduce the spark intensity trend even more, leading to a more compact microstructure and sometimes higher growth rate [28–32]. In one aspect, the oxide coating behaves like a capacitor during anodic and cathodic polarizations, causing the *I–V* of two pulses connected. The preceding anodic pulse provides remnant positive charges, which diminish the probability of cathodic discharges in the following negative polarization. Furthermore, a combination of unipolar and bipolar modes, satisfying different needs of AM60B at different stages, may generate the superior microstructure for corrosion protection [33]. This result implies the electricity waveform ought to vary with the micro arc state in the pursuit of optimal coating.

In principle, PEO is an oxidation process, it seems that the surface treatment requires positive (anodic) current only. Yet the involvement of cathodic current plays a critical role in controlling the coating defects and the coating response to anodic current. During growth, addition of the cathodic current raises the number of point defects, alters the coating resistivity, such that the rising trend of coating resistance diminishes. At the right moment, decline of the coating resistance alleviates the danger of destructive discharges, type B discharges. Furthermore, the influences of cathodic current accumulate over a longer time frame, the entire operation period. Soft sparking transition is an iconic phenomenon; the voltage suddenly decreases at a later time in stage (IV). The voltage drop signifies an evident shift in coating resistivity due to accumulating differential changes in its electronic or ionic conductivity.

The long-term effects of cathodic current can be found in current–voltage curves (CVCs) and voltammograms. Figure 2 presents two schematic voltammograms of milliseconds duration, contrasting the differences with and without cathodic current [34]. The voltammetric plots can be classified into two kinds; Figure 2B without the cathodic current influences, Figure 2C with the cathodic current influences. Figure 2B depicts two exponential-like current curves in the increasing and decreasing scans. This diagram displays the CV scan current of a double-layer capacitor with some degree of current leakage, marking the features of PEO coating without cathodic current involvement. Figure 2C indicates a bulge in the ascending scan, no bulge in the descending scan. The bulge is the feature of PEO coating with cathodic current higher than anodic current, after an erase of short-term remnant charges. When the *I–V* data of Figure 2C are redrawn as CVC of Figure 2D, there is also a bulge in CVC. We realize the bulge is not a valve effect either, since, at particular point, the d*I*/d*V* value is negative. In real operations, the bulge is tiny in the early PEO, growing with the PEO time. Hence the bulge feature is most evident near the end, a similar difference can be found in comparing CVCs of Figure 2A,D in the electrolyte of Na$^+$ and K$^+$, using 50 Hz AC [12]. The bulge arises from accumulation of the so-called "cathodic induced changes" (CIC). CIC is an imprint of cathodic current on coating, being interpreted as extra hydrogen has been absorbed in coating and accumulated as the Frenkel defects over time [34].

**A.** Voltammetric input
for diagnosis

**B.** Charging double
layer of interface

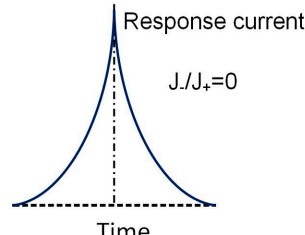

**C.** Charging the interface
with accumulated CIC

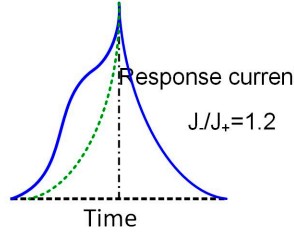

**D.** CVCs with and without
accumulated CIC

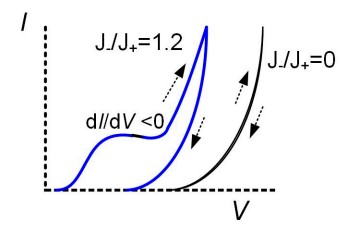

**Figure 2.** Schematic diagrams showing the effects of cathodic current on the response of voltammetric diagnosis input. Reprinted with permission from [34]; Copyright 2019 Elsevier. The schematic diagrams show (**A**) the voltammetric diagnosis input pulse of two constant scan rates of ascending and descending voltage; (**B**) the response voltammetric current of a plasma electrolytic oxidation (PEO) coating prepared by zero cathodic current; (**C**) the response voltammetric current of a PEO coating prepared by negative current higher than positive current; (**D**) the current–voltage response curves (CVCs) of two PEO coatings prepared with and without cathodic current.

## 4. Growth Species and Paths

### 4.1. Growth Path 1

Over the years, researchers have increased the electrolyte content to provide the PEO coating more functionalities, which also affects the growth behavior and dielectric properties. We may classify the growth species and their paths into five categories, as shown in Figure 3. Path 1 is the inward growth in which hydroxyl ion, $OH^-$, diffuses through the coating and oxidizes the metal [35]. This path of diffusion and oxidation is ubiquitous, producing a dense layer of native oxide with high adhesion strength. This dense layer is critical to the quality of protection. In case of aluminum, the native oxide layer is a mix of defective boehmite and gamma alumina. On its path toward metal, hydroxyl ion may react with outward-bound $Al^{3+}$ and proceed dehydrogenation simultaneously. The contribution of $Al^{3+}$ counter-diffusion is considered small on growth, compared with that of hydroxyl ion, since $Al^{3+}$ diffusivity is much less than that of $OH^-$ and its driving cathodic voltage is relatively small. Hydrogen shall come from interfacial reactions between Al metal and $OH^-$. Hydrogen ought to be able to escape with ease, since its size is very small. Still, as the layer thickness increases, the inward growth rate decreases because the diffusion distance increases and the electric field declines, known as self-limiting growth in anodizing [36]. Although suffering from the same restriction, the inward growth of PEO is much more evident than that of anodizing due to high voltage.

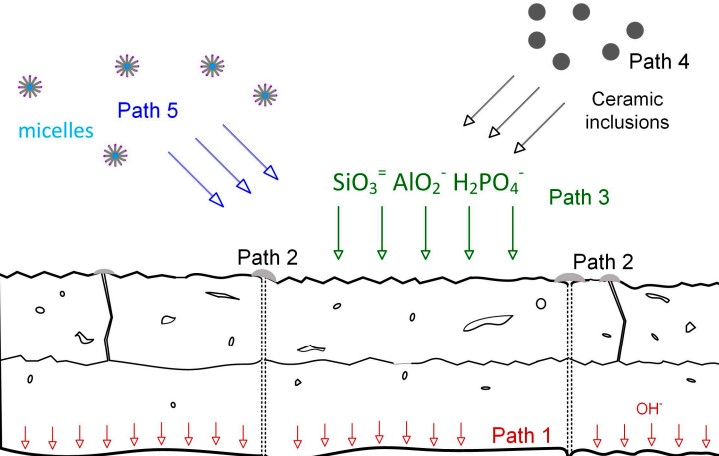

**Figure 3.** Schematic diagram of five growth paths and their growing species. Path 1 is the path of hydroxyl ion oxidizing the metal substrate; path 2 involves the deep and shallow discharges melting oxide and freezing again; path 3 is the oxyanions adsorption and deposition; path 4 is the incorporation of ceramic particles; path 5 involves organic polymeric colloids or dye micelles incorporation.

In the situation of magnesium, the product of path 1 is not ideal, since the MgO layer is highly alkaline and hygroscopic, vulnerable to acid attack. Insufficient protection of this native oxide is intrinsic to the low corrosion resistance of Mg-alloys. To upgrade its anticorrosion capability, many researchers resort to the fluorine precursors and establish more corrosion resistant $MgF_2$ layer. The solubility of $MgF_2$ is moderate, $2.6 \times 10^{-4}$ M at room temperature, and expected to increase at low pH. Still, $MgF_2$ is superior to MgO in anticorrosion by a substantial margin. The coatings containing $MgF_2$ do show improvement in corrosion protection [37–39]. However, $MgF_2$ cytotoxicity is of concern when the coating is applied in biomedical materials [40,41]. The other intriguing fact of fluorine involvement is these electrolye solutions are often acidic [37,38]. Fluorine of the electrolyte reacts with Mg metal during the initial contact, and persists at the coating/metal interface when driven by the high voltage of PEO. Details of diffusion mechanism are not known; either fluorine ion diffuses with oxygen ion into metal, or the fluoride-rich layer holds back the later diffusion of $OH^-$. Either way, $MgF_2$ and MgO coexist at the oxide/metal interface [42,43].

In the situation of Ti-alloys, $OH^-$ again plays a significant role, the product is a dense native oxide layer of anatase phase [44]. This dense barrier layer is between the porous coating and the metal substrate. Right above the dense layer, beneath the porous layer, there are clusters of pore, which are exclusive to Ti-alloys, not found in Al- and Mg-alloys. These pore clusters are presumably generated due to phase transition of anatase ($3.89$ g·cm$^{-3}$) to rutile ($4.25$ g·cm$^{-3}$) [44–46]. Titanium samples may be PEO treated in sulfuric and phosphoric acids [47]. Its interface issue of PEO has been studied in 1.5 M sulfuric acid, sulfur is incorporated between oxide coating and titanium metal [48]. The sulfur presence at the oxide/metal interface appears similar to the fluorine of Mg-alloy coating.

### 4.2. Growth Path 2

Path 2 reshapes the coating microstructure, more than adds on its thickness. When the coating reaches a substantial thickness, the electric discharge could occur at the top of the coating or within the pores. Th electron temperature $T_e$ of the discharge is very high. According to Clyne and coworkers [49], the core temperature was estimated $16,000 \pm 3500$ K, and the peripheral temperature was ~3500 K. The average value of $T_e$ increases with PEO time, then decreases [50,51]. For the high-spike discharge, the individual $T_e$ value generally increases with PEO time. These electron temperatures are based on the kinetic energy of electron, not the thermodynamic temperature that we are familiar with. Meanwhile, there is no direct conversion between electron and thermodynamic temperatures. Nonetheless the work of Lee and coworkers [52] provided a glimpse of their connection in PEO. They

studied the PEO treatment of Mg-alloy, indicating that $TiO_2$ inclusion melted, while $ZrO_2$ inclusion did not. In other words, the thermodynamic temperature of plasma in stage (IV) could be deduced between 2116 and 2643 K. The bottom line was that discharges were sufficiently hot to melt the nearby Al- or Mg-oxide or even the substrate metal.

The melt of metal or oxide enters the discharge channel, goes upward and encounters the oxyanions to react and cool off. In this manner, the melt regenerates the outer surface and enhances mass transfer vertically. The terrain features of path 2, micron-sized frozen lava and craters of the discharge channel, are eye-catching under electron microscopes. Cracks often stem from the discharge channel due to thermal stresses. On the Al-alloys, these features are manifested in the results of Curran and Clyne [53] at 100 mA·cm$^{-2}$ of constant current mode, and in those of Cheng and coauthors, exceeding 250 mA·cm$^{-2}$ [54]. Similar morphological features are also evident on Mg-alloys when current density is over 50 mA·cm$^{-2}$. For example, Hussain imposed a positive current of 70 mA·cm$^{-2}$ on AJ62 Mg-alloy, and the coating surface was rich in frozen lava [55], in the Zhuang's work on AZ31 [56]. Either 150 mA·cm$^{-2}$ [57] or 250 mA·cm$^{-2}$ [54], when the current density exceeds the typical range for Al- and Mg-alloys, the features of path 2 are distinctive. In case of Al-alloys, if the $\alpha Al_2O_3$ phase is desired, the operation near 100 mA·cm$^{-2}$ with sufficient long duration is recommended, since joule heating assists in crystal growth of $\alpha$-phase nuclei.

*4.3. Growth Path 3*

The common oxyanions of path 3 are $SiO_3{}^{2-}$, $AlO_2{}^-$, $PO_3{}^-$, $H_2PO_4{}^-$, $HPO_4{}^{2-}$, $P_2O_7{}^{4-}$, $B_4O_7{}^{2-}$, $VO_3{}^-$, $WO_4{}^{2-}$, $MnO_4{}^-$ in form of electrolyte additives $Na_2SiO_3$, $NaAlO_2$, $(NaPO_3)_6$, $KH_2PO_4$, $K_2HPO_4$, $Na_4P_2O_7$, $Na_2B_4O_7$, $NH_4VO_3$, $Na_2WO_4$, $KMnO_4$ [58–62]. Some of them render colors to the coating. For instance, $VO_3{}^-$ and $WO_4{}^{2-}$ [63] furnish the black color. Most of them are also known as passivating anions, since they passivate the metal surface and diminish the anodic dissolution of metal. Silicates, vanadates, tungstates, and mangnates have good passivation effects. The other effect, seldom mentioned, is the effect of diluting the valve effect of Al- and Mg-alloys. Incorporation of most oxyanions in coating diminishes the valve effect, except $AlO_2{}^-$.

Perhaps the most important anion of passivation is $SiO_3{}^{2-}$, whose incorporation is known to make the coating even and thick [28]. $Na_2SiO_3$ is highly soluble and inexpensive. However, too much electrolyte silicate creates high porosity [64], hence its content seldom exceeds 10 g·dm$^{-3}$. The usual $Na_2SiO_3$ concentration range is 1–5 g·dm$^{-3}$ in electrolyte. Silicate anion reacts with the metal oxide, and yields noncrystalline glass, or aluminosilicate crystals. Mullite $3Al_2O_3 \cdot 2SiO_2$ is typical in the coatings of Al-alloy PEO, while the $Mg_2SiO_4$, $MgSiO_3$ compounds are often found on Mg-alloys. The importance of $SiO_3{}^{2-}$ is exemplified in a study of developing the so-called "universal" electrolyte. The idea of universal electrolyte originates from the necessity of industrial PEO, since replacing a large volume of electrolyte is not cost-effective for industrial operations. Hence, an electrolyte recipe applicable for both Al- and Mg-alloys is investigated. Kossenko and Zinigrad [65] chose two ingredients only, KOH and $Na_2SiO_3$, in studying the universal electrolyte. Phosphate anions play a similar role as silicate anion, since they both are glass formers. Yet metal silicates are generally more refractory and corrosion resistant than metal phosphates. Therefore, the coating thickness increases faster in silicate electrolytes, than phosphate electrolytes [28]. When the $Na_2SiO_3$ concentration is high, silicate anions allegedly polymerize into colloids or clusters, and effect an inorganic polymer gel [66]. These macromolecules may preserve the joule heat and assist densification.

Sodium aluminate $NaAlO_2$ is quite soluble in alkaline solutions. $NaAlO_2$ concentration may reach 56 g·dm$^{-3}$ in electrolyte [28,54]. The solution of $NaAlO_2$ contains a number of oxyanions, existing in form of complex clusters of chain anion and ring anion. The simplest is known as $AlO_4{}^{5-}$ [67]. A combination of $NaAlO_2$ and sodium benzoate has been reported to enhance the corrosion resistance and the $\alpha Al_2O_3$ content of coating on the 6061 substrate [68]. In PEO of Mg-alloys, $NaAlO_2$ in the electrolyte reacts with MgO, and

yields spinel $MgAl_2O_4$, which is an iconic phase for the coating of superior corrosion and wear resistances.

The other two additives, $K_2ZrF_6$ and $Na_3AlF_6$, provide $ZrF_6^{2-}$ and $AlF_6^{3-}$ fluoroanions. They are valuable resources for $MgF_2$, $ZrO_2$, $MgAl_2O_4$, and $Al_2O_3$ in the PEO coatings of Mg-alloys. Acidic $K_2ZrF_6$ is typically introduced, using a two-step procedure [38,69,70]; first establishing a corrosion resistant layer of rectification in electrolyte I, then PEO in the electrolyte II of $K_2ZrF_6$. The electrolyte I usually does not contain $K_2ZrF_6$, may contain fluorine. The pre-coated layer facilitates the uniform micro arc of the second step. $Na_3AlF_6$ has been added into the electrolyte for PEO Al-alloys. A proper amount of $Na_3AlF_6$ is shown to enhance the coating thickness and reduce the pore size of coating [71,72]. Even though fluorine can be convenient for pore elimination, the fluorine content of PEO coating is always a health concern.

*4.4. Growth Path 4*

Suspended particles can be incorporated, and contribute to the growth rate. Path 4 denotes the route in which inorganic particles are entrapped in the effluent of discharge and return to the coating surface to deposit. Path 4 is established on the observation that uptake particles have been congregated on the surface area of frozen lava around a crater [73]. This piece of observation does not exclude the probability of direct inclusion without discharges. The particles are added intentionally, not those nanoparticles that have been self-generated in plasma [74]. The foreign particles of nanometer size are intrinsically agglomerated in aqueous solutions, hence deflocculation is necessary for a uniform coating. The simplest way of deflocculation is to adjust the pH value. For example, many oxides and their hydroxides have an isoelectric point below 9, such as $Al_2O_3$, $SiO_2$, $MnO_2$, $TiO_2$, $CeO_2$, $ZrO_2$, $SnO_2$, $WO_3$ [75]. These particles are negatively charged and the electrostatic repulsion between particles can be enhanced through adjusting pH. Hence, they are incorporated during positive polarizations. Isoelectric points of a few oxides are above 9–10; for instance, NiO, CuO, PbO, MgO, ZnO, $La_2O_3$, they are rarely chosen as inclusions for PEO. If chosen, addition of a commercial dispersant, such as Darvan C–N, improves the suspension stability and inclusion probability. Darvan C–N is an anionic dispersing agent often used to disperse the ceramic powders without causing foams. For non-oxides, their isoelectric points ought to be experimentally measured, since their surfaces are often covered with oxygen or hydroxide, and the oxygen coverage affects the isoelectric point. If insufficiently dispersed in aqueous solutions, the surface of non-oxides can again be modified with various anionic dispersants. The dose of commercial dispersants is usually less than 3%, with respect to powder weight, sufficient for deflocculation. The electrolyte dispersant is necessary in Mg-alloy PEO, since Mg-metal is etched in presence of hydroxide. The inclusion particle surface is often covered with magnesium hydroxide and with high isoelectric point. Researchers have studied their suspension stability issues, and concluded that dispersants of low-molecular weight, such as SDS and urea, are preferred [76].

The strong suit of path 4 is its flexibility. A wide range of particle precursors, either oxides or non-oxides, can be chosen to modify the coating. The weakness of path 4 is the often-insecure binding between inclusion and matrix. Non-oxide particles, such as $Si_3N_4$ [77,78], SiC [79,80], TiN [81,82], TiC [83,84], have been incorporated in PEO to improve hardness or wear resistance of the Mg and Ti-alloy surfaces. On the Al-alloy surface, the incentive to incorporate nitride or carbide is not strong, since in-situ crystallization of $\alpha$-alumina phase can render sufficient hardness and wear resistance to the coating. In principle, nucleation of $\alpha$-alumina is probable in the Mg-alloy PEO with a sufficient $NaAlO_2$ content in the electrolyte. But, to the best of our knowledge, no work has been done on in-situ crystallization of $\alpha$-$Al_2O_3$ on Mg-alloys. Addition of $\alpha Al_2O_3$ [85–87], $ZrO_2$ [88–90], $CeO_2$ [91–93] nanoparticles is a quick and energy-saving way to increase wear and corrosion resistances of Al-, Ti-, and Mg-alloys. The electrolyte additive $Na_2SiO_3$ is often accompanied with the preceding oxides to assist bonding between nanoparticles and coating matrix. Clay powders are another inexpensive source for silicates of the PEO

coating. Most of clays have low melting temperatures, which may explain why they are incorporated to enhance the corrosion resistance of Mg-alloy coatings [94–96]. When certain applications demand a low surface friction, $MoS_2$ [97–99], graphite [100] particles are the common inclusions.

Inclusion particles are normally found on the upper porous layer of the coating, not in the dense inner layer. Merely physical trapping in the pores cannot secure these nanoparticles. On the other hand, establishing the solid bonding requires a trade-off between intended properties and solid-state oxidation. Many applications demand substantial solid bonding, which involves solid-state reaction between particle and surrounding matrix [101]. Nonetheless, solid-state reaction can diminish the intended properties of particles. Solid-state reaction between oxides and nitrides results in oxynitrides and oxides. In PEO of MA8, TiN particles were oxidized into a mixture of $TiO_xN_y$ oxynitride and $TiO_2$ [82]. In PEO of AZ91, SiC nanoparticles were partially turned into silicon dioxide [79]. It is difficult to image non-oxide particles escape the hot discharges, react with its surrounding oxide, and remain intact under such a strongly oxidizing environment.

In terms of solid bonding, $SiO_2$ is perhaps the best example, since, when melts, it forms various oxide glasses with alkaline cations. As the glassy phase is introduced in the micro arc state, liquid phase sintering also enhances densification of the porous coating. Lu and coworkers have studied the size effect, using nanometer $SiO_2$ particles (12 nm) and micrometer-size particles (1–5 μm) [102]. They concluded that $SiO_2$ particles of 12 nm were reactively incorporated, solid bonded, while the micrometer-size particles were inert to the coating matrix. They inferred the coating of reactively included $SiO_2$ was suitable for bio-medical applications, suggesting reactive incorporation was superior in the coating strength.

Inclusion of $Cr_2O_3$ poses a peculiar case, in which electrochemical rectification is severely interfered by its inclusion. In the inclusion studies [103,104], the electrolyte contained 0.375 g·dm$^{-3}$ NaOH, 4 g·dm$^{-3}$ $NaAlO_2$, and 10 g·dm$^{-3}$ $Cr_2O_3$ powder, whose average particle size was either 351 or 68 nm. Hence, this specific growth involved path 1, path 3, and path 4. The electrochemical oxidation of Al-alloy 6061 was performed using a pulsed current of bipolar waveform with negative current 92 mA·cm$^{-2}$ and a positive current slightly higher or less than 92 mA·cm$^{-2}$ in a constant current mode or constant voltage mode. The $Cr_2O_3$ crystal is $p$-type, just opposite to the $n$-type behavior of $Al_2O_3$/Al interface. When $Cr_2O_3$ was included in the coating growth, the voltage-time profile displayed two very different features. The first noticeable difference was that, instead of a smooth transition, the voltage surged as the process went from stage (III) to stage (IV), then returned to the slowly increasing trend, Figure 4a. The voltage surge was found in both positive and negative voltage-time profiles. The second marked difference was the voltage might escalate and run away when the operation entered stage (IV) after a period of time, depending on the magnitude of positive current. When the positive current density exceeded 92 mA·cm$^{-2}$ by a substantial quantity, the positive voltage would run away. The two features suggested that the inclusion of $Cr_2O_3$ plus $AlO_2^-$ raised the coating resistance, and it ran away at certain percolation point. Tsai and coauthors [104] assumed that solid-state reaction occurred between the $Cr_2O_3$ inclusion and the depositing aluminum oxide, and the reaction produced a depletion layer, a junction where $p$-type and $n$-type carriers recombine. The depletion layer was not conducting, hence, producing extra resistance and promoting the local discharges. When the depletion layer was a minor phase, the voltage progressed as usual. When the depletion layer content reached a percolation point, the voltage surged to an unreasonable level, the discharge might blow away a local piece of $Cr_2O_3$/$Al_2O_3$, as shown in Figure 4b. The voltage surge could be so high that the PEO treatment was forced to shut down to safeguard the power supply. More discharges occurred when the particles of small size were incorporated. In case of $Cr_2O_3$, these interior discharges of 68 nm $Cr_2O_3$ inclusion created more porosity than 351 nm $Cr_2O_3$, since small size inclusions went deeper in the coating.

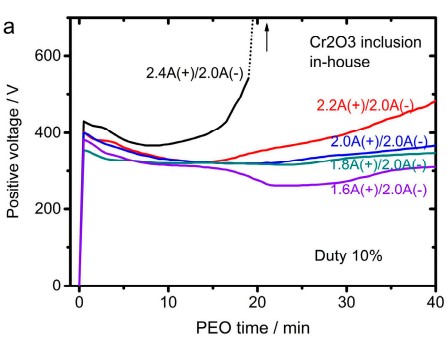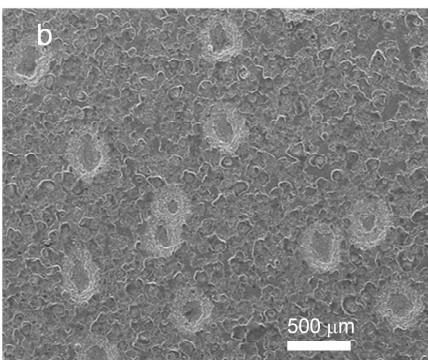

**Figure 4.** PEO with $Cr_2O_3$ inclusions. Reprinted with permission from [103,104]; Copyright 2017 Elsevier, Copyright 2017 American Chemical Society. The (**a**) voltage profiles of constant current mode using pulsed current of bipolar waveform with various positive current settings; 2.4 A(+)/2.0 A(−) denotes positive current 2.4 A and negative current 2.0 A for the 6061 sample of 21.7 $cm^2$. Please note the positive voltage run away around 19.5 min. A typical (**b**) surface image of the coating sample, note that several small pieces of the coating are blown away, creating shallow dents of white color. The discharges, due to depletion layer at the $Cr_2O_3/Al_2O_3$ interface, are violent, causing these shallow dents.

$K_2ZrF_6$ may contribute to the coating via path 3 or path 4, depending on pH of the electrolyte. The colloidal aspect of $K_2ZrF_6$, path 4, is often ignored in the PEO literature. Many researchers assume the precursor is in the molecular state, as long as the solution remains visually transparent. When the pH value of electrolytic solution is between 10 and 12, the colloidal solution of $K_2ZrF_6$ is well dispersed due to electrostatic repulsion, since the isoelectric point of zirconium hydroxide is 6.7 [75]. If the particle is sufficiently small and not agglomerated, the colloid could appear transparent. At ambient temperature, $K_2ZrF_6$ solubility is moderate in acidic solution, ~18 $g \cdot dm^{-3}$ (pH < 5), dissociated into $ZrF_6^{2-}$ and $K^+$. In neutral or alkaline solutions, $K_2ZrF_6$ is hydrolyzed into negatively charged particles of $Zr(OH)_4$ (6 < pH < 10), or $Zr(OH)_5^-$ (pH > 11) [105].

*4.5. Growth Path 5*

Path 5 may be viewed as a variant of path 4, since the inclusions are polymers or organic molecules, instead of inorganic particles. This composite of oxides and polymers is also referred as duplex coating. The most widely-studied duplex coating is a coating of PEO and PTFE on Mg-alloys, since PTFE upgrades corrosion resistance, reduces surface friction, and adds surface hydrophobicity. There are two approaches to realize the duplex coating. In the two-step procedure, the metal surface was oxidized in PEO treatment first, then a colloidal PTFE solution was applied to fill the coating pores under vacuum [106,107]. Tsai and coworkers pointed out that pore filling was critical to the anticorrosion properties [18]. The PTFE-based precursor performed better than a commercial PTFE colloidal solution, since it was imbibed at a less viscous state, then cured to harden the precursor and seal the pores tightly. In the one-step procedure, suspended PTFE particles were incorporated via path 5 during PEO. A portion of PTFE was decomposed in the surface plasma. The question is how to minimize the extent of decomposition. In that sense, the constant voltage mode is more adequate than the constant current mode. Moreover, the current, under constant voltage mode, was lower with PTFE than that without PTFE except at the end of PEO treatment [108]. Evidently, the insulating nature of PTFE shall diminish rectification of the PEO coating. Near the end, the discharge damages are expected since the surface coverage of PTFE is nearly full. A high level of PTFE degradation is expected when the constant current mode is imposed, since PTFE has to endure high voltage for a long period of time. A duplex coating example of in-situ incorporation was given on PEO of Al-alloy, in which PTFE inclusion was accompanied with siloxane-acrylate emulsion [109]. The resultant

coating was reported to exhibit the improved properties of hydrophobicity, anti-friction and anticorrosion, even when a significant fraction of PTFE inclusion decomposed.

Synthetic dyes are organic molecules with more than one chromophore, which contains a molecular structure of alternating double and single bonds. Tsai and coworkers colored the PEO surface with the technique of in-situ incorporation of organic dye emulsions. This technique produced a vivid coloring effect, Figure 5. The technique was not restricted to a specific dye, more uniform than that of the two-step procedure of coloring after regular PEO [110]. These dyes were introduced into the electrolyte in form of micro-emulsions, which was included via path 5 under the constant voltage mode. In these experiments, dyes seemed to be subject to less damages than PTFE in the PEO plasma. In principle, dye molecules are not more durable than the polymeric PTFE. We postulate that those low-weight molecules never enter the discharge envelope in their path toward inclusion. Another unique aspect of dye inclusion is that these organic dyes diminished the discharge activity significantly. The current dropped so drastically that the distinct plasma extinguished in a short period of time. In an effort to revive the plasma activity, the voltage was raised in a series of manual action to maintain the uptake. Consequently, a coating of dye and oxide mixture was obtained with sufficient thickness.

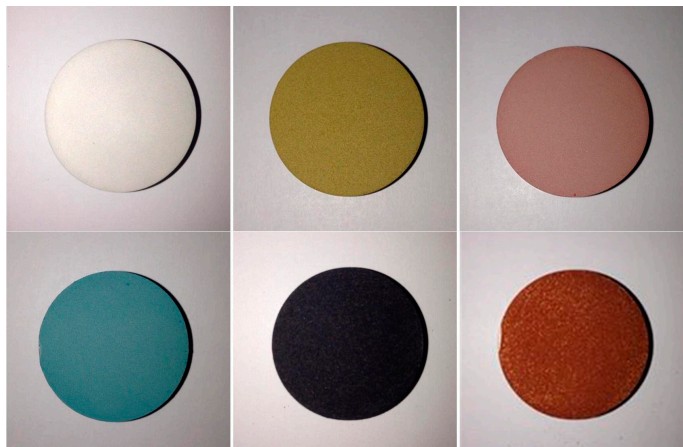

**Figure 5.** Colorful PEO coatings on the 6061 Al-alloy sample. Reprinted with permission from [110]; Copyright 2016 Elsevier. These colors have been created through in-situ incorporation of several organic dyes under constant voltage mode via path 5 in form of direct micro-emulsion.

## 5. Microdischarges and Their Control

### 5.1. Electric Discharges and Physical Quantity Denoting Micro Arc State

As the coating increases in thickness, so does the spark intensity in general. However, not every discharge event overcomes the same breakdown barrier. In reality, the breakdown strength at the interface is widely distributed, so is the individual spark intensity [111]. A high-spike breakdown in the pore is the focus of attention. When the pore geometry is deep and narrow, this pore geometry may provoke a cascade of discharges at the same location [1,112]. Normally, this type of local discharge cascades ought to be avoided, since they generate a weak spot in microstructure. Early in stage (IV), the entire surface is wrapped up in well dispersed sparks which are large in number and tiny in size. Hence, the probability is very low for a combination of deep pore position and locally persistent discharges. If oxidation discontinues at that moment, the coating quality is excellent. However, the coating thickness is often insufficient. As oxidation proceeds, this favorable state of micro arcs deteriorates. The size and intensity of spark increase with time, and its spatial number decreases, the probability of damaging spark increases [113,114]. Given the adverse trend, we ought to develop the techniques of monitoring the micro arc state to know when and how to regulate the electrical parameters. To this end, researchers have

been searching for a physical quantity that diagnoses the micro arc state for the on-line control purpose.

*5.2. OES as Diagnostic Tool*

Many researchers view the emission lines of optical emission spectroscopy (OES) contain such a physical quantity [115]. A typical OES setup employs a CCD array detector coupled with an optical fiber to focus and scan the sample surface and collect optical emissions in millisecond. If the sample area is sufficiently small, the collected data shall originate from a single discharge event or a cluster of discharges (cascades) [48,116,117]. The study of individual discharge reveals the nature of breakdown events. Typical electron temperature $T_e$ of local discharges has been estimated [48]. When the entire sample surface of large area is scanned, the signal is collective, denoting the micro arc state of entire sample. Values of electron temperature have been calculated from the global emission spectrum and used in plasma analysis [118,119].

Nonetheless, the meaning of OES signal may not be straightforward. As pointed out by Clyne [48], the operating voltage remains approximately constant as the coating grows in stage (IV), implying the electric field required for dielectric breakdown is approximately the same regardless the thickness increase. Clyne and coauthors postulated the breakdown events could occur within the porous oxide structure near barrier layer. This conjecture suggests many breakdown events are hidden in pores of the coating, could be as deep as the oxide/metal interface. It may not always be true for all PEO conditions. Nevertheless, the postulate points out the most damaging discharge could be the most obscured emissions, since its root could be deep down at the oxide/metal interface.

Three types of discharges have been outlined [115,118]. Type A discharges are breakdown events at the electrolyte/oxide interface; type C discharges are events in the pores, occurring at a position between two interfaces of electrolyte/oxide and oxide/metal. Discharges of type A and C are weak in intensity. Type B discharges are strong events, occurring near the oxide/metal interface. If electron temperature is the target to control, we ought to have the capability to draw a line, with $T_e$, between the group of type A and C and the group of type B. Many profiles of $T_e$-time, featured with significant fluctuations in magnitude, have been documented. To the best of our knowledge, those spikes of type B are never singled out with justifiable criteria. If type B discharges cannot be identified, no control strategy can be formulated. The issue of choosing a proper element is equally difficult to answer. Researchers have monitored the emissions of several elements over a long period of processing time. Their results indicate the intensity of Al lines decreases with time from the beginning, the intensity of sodium lines increases with time. The $H_\alpha$ line does not vary much with time, yet the hydrogen line mostly related to water electrolysis [120]. Does the individual intensity of $H_\alpha$ line denote the intensity of breakdown? Although Hermanns and coauthors have recently shown the feasibility of in-situ PID control through image processing of spark size with a unipolar pulsed current [121], their control strategy is based on a precarious connection between spark size and damaging discharge. Besides, the electricity waveform nowadays is far more complicated than the unipolar pulsed current.

In our view, the greatest triumph of OES is to prove that cathodic current of bipolar pulsed current cools down the uprising trend toward high-spike discharge, especially when we compare the pulsed current of bipolar waveform against unipolar waveform. Martin and coworkers [31] have shown the cool-down effect of negative current on the high spikes in two consecutive cyclic pulses. Hussain and coauthors [122–124] have demonstrated, via $T_e$ analysis, the superiority of bipolar mode on the resultant coating microstructure, compared with the unipolar mode. In addition, they have shown the cathodic current of bipolar mode benefits the tribological properties of coated AM60B [125].

*5.3. Dielectric Properties as Diagnostic Tools*

The other physical quantity may come from the transient analysis, which provides the information about dielectric properties of the interface. Researchers have analyzed

the transient response of bipolar pulsed currents and tried to find the equivalent circuit terms denoting the micro arc state. Fatkullin and coauthors [126–128] were the first to take this approach using the constant voltage mode, and proposed an equivalent circuit to fit the transient response. Three transients, two in the current and one in the voltage, were fitted with an equivalent circuit. The earlier results are disappointing, since only one correlation of a particular resistance or the sum of two resistances was shown to follow the monotonically increasing trend of coating thickness. Using the constant voltage settings, analyzed with a more sophisticated equivalent circuit, the second report provided more physically meaningful time evolutions of $R_1$-$C_1$ and $R_2$-$C_2$, which were suggested to be resistance and capacitance values of the porous upper layer and the dense inner layer; respectively. The values of $R_1$, $R_2$, $C_2$, increased, $C_1$ decreased with PEO time. The reason of diminishing $C_1$ was not clear. There were also unexplained trends in the fitted parameters of inductance $L$, and other resistances $R_3$, $R_4$.

A better platform of diagnosis is soft sparking transition, since the sudden drop in voltage, 15%–40%, affects all aspects of the micro arc state. If the proposed physical quantity can capture this dramatic change, that is, going up or down with the transition, this quantity is tracking the micro arc state. Gebarowski and Pietrzyk [129] attempted to find soft-sparking like transitions in the evolution of equivalent circuit parameters with progressing time. They took out the PEO samples, after 10, 20, 30, 40, 50, 60 min treatment, and measured their impedances, fitted the impedance results with a model of resistor and constant-phase element (*CPE*) in series. The cell voltage dropped between 30 and 40 min, so did $R_1$ and $R_2$. However, variations in $R_3$, $CPE_1$, $CPE_2$, and $CPE_3$ values did not show abrupt changes. Thus, there could be one quantity or two, which grasped the essence of micro arc state. As a whole, the technique is not suitable for on-line control, since no one shall interrupt the PEO treatment to perform impedance measurements.

Additionally, on the soft-sparking platform, Tsai and Chen [130] examined transients of the bipolar pulsed current of galvanostatic PEO in the electrolyte of 5 g dm$^{-3}$ Na$_2$SiO$_3$ and 1 g·dm$^{-3}$ KOH. Both transient current and voltage values were measured in-situ with a dual-channel oscilloscope at an interval of two minutes. The transient behavior of the positive pulse was chosen to be correlated, not the negative pulse, since the residual charges of the positive pulse could not be eliminated entirely during the recess period between positive and negative pulses, and the residual charges interfered the transient behavior of negative pulse. The influences of residual charges were more and more significant as coating thickness increased, because the capacity of interface capacitor increased with increasing thickness. On the other hand, the negative pulse did not leave remnant charges to interfere the transient of positive pulse because of valve effects. The proposed equivalent circuit, two *RC* elements in parallel with a current leakage element, was capable of fitting the transient current with the coefficient of determination $R^2$, higher than 92%, except an initial period of considerable fluctuation.

The correlated values of time constants (*RCs*) could track the variations in soft sparking transition at 50 Hz with various combinations of positive and negative current settings. Furthermore, the time constants, $R_1C_1$ and $R_2C_2$, declined abruptly before the cell voltage dropped. In other words, time constants of the transient were more sensitive to soft sparking transition than the power supply controller, which detected a lower voltage was sufficient, via feedback loop, to maintain the preset value of current. The above conclusions were verified in two electrolytes; a conventional electrolyte of KOH and sodium silicate, also in the solution of identical recipe with chromia inclusions. The soft sparking transition of 500 Hz occurred prior to those of 100 and 50 Hz, and the time lag between voltage drop and $R_1C_1$ drop became very small at 500, since 500 Hz is much like nonstop AC. Meanwhile, the large time constant $R_2C_2$ could be associated the coating microstructure at 500 Hz. For unknown reasons, at 500 Hz, 40% duty, 1.2 A(+)/1.5 A(−), fluctuations of $R_2C_2$ with respect to processing time appeared to coincide with the stratified layer structure. A similar stratified layer of more subtle microstructure was also reported in [131], in which the PEO treatment was operated in a soft-sparking condition using an electrolyte of

1.65 g·dm$^{-3}$ Na$_2$SiO$_3$ and 1 g·dm$^{-3}$ KOH. A lamellar structure of 1:1 mullite/alumina was formed during soft sparking transition, and interpreted as the result of phase separation via spinodal decomposition [128].

*5.4. Cathodically Induced Change*

Yang and coworkers first proposed a waveform which consisted of a diagnostic pulse regime and a potentiostatic work regime [132]. The potentiostatic regime involved positive polarizations only, and the authors concluded that the impedance evolved in a trend complying with the coating growth. They further programmed a self-adaptive control loop, and utilized a set of criteria to control the positive voltage according to the diagnostic pulse feedback [133]. Unfortunately, they did not understand the context of soft sparking at that particular moment, since they claimed electric discharges were in the soft sparking regime without cathodic current involvement.

A more sophisticated diagnosis method was proposed recently by Rogov, Matthews, Yerokhin [33,134]. Its effectiveness was also demonstrated on the platform of soft sparking transition. Their pulse waveform was designed with a pulse train of working and diagnostic segments. An in-house power supply provided a one-second working pulse train of 50 Hz bipolar pulsed current with the current ratio 1.2 or 1.3 ($J_-/J_+$ or $Q_-/Q_+$), followed by a diagnostic pulse train of one-second duration with three consecutive pulses of extracting pulse, injection pulse, and potentiodynamic (voltammetric) diagnosis pulse. The extracting pulse was designed to erase the residual charges of preceding pulse train. The injecting pulse was a galvanostatic pulse designed for erasing other charges, here, 4.06 milliCoulomb. The fixed amount of charges was injected to balance the charges difference between ascending voltage scan and descending scan in diagnosis pulse. The voltammetric pulse was a positive triangle with an equal scan rate of ascending and descending voltage. Since the cathodic charge ($Q_-$) was 30% higher than the anodic charge ($Q_+$), the negative current of working pulse train accumulated permanent changes in coating like an imprint, the current response in voltammetry pulse was not rising and falling at the same rate. The distortion in current could be analyzed to extract the CIC. They derived the "hysteresis charge" in analyzing CIC. The term "hysteresis charge" was shown to go up and down, coinciding with the drops of light emission and voltage in soft sparking transition. The authors viewed the physical quantity of hysteresis charge as the degree of oxide protonation, and formulated a linear differential equation to describe its kinetics over the entire stage (IV). In short, the authors assumed that an oxide region of coating underwent protonation and deprotonation processes, of which the kinetics could be extracted from the diagnostic pulse. The amassed protonation affects the coating conductivity. The works of Rogov and coauthors give a concrete meaning to the physical quantity "hysteresis charge" which is responsible for the coating resistance drop during soft sparking transition. Meanwhile their works also rouse more questions. It seems the oxide region undergoing protonation and deprotonation is a part of dense inner layer, which may be thin, thick, or varying in thickness. Its decisive role requires more justification. Additionally, it is not clear how to decide the charge quantity to inject such that the permanent changes are isolated from temporary changes during accumulation. It seems unreasonable to have a fixed charge quantity in injection, because the coating is growing, so is the retained quantity. Furthermore, the assumption of protonating oxide region, governing conductivity of the entire coating, seems to oversimplify the behavior of a growing dielectric system [33]. Nonetheless, this kinetic model has paved the way to control the pulsed current waveform in a physically meaningful way. It is a huge achievement toward the insightful control.

## 6. Conclusions and Future Outlook

We have categorized various contributions to the PEO coating into five growth pathways. Path 1 involves OH$^-$ diffusion through the coating, and establishes a dense inner layer adjacent to the substrate metal. Path 2 involves a discharge event melts the inner matter and carries to the surface, cools and reshapes its coating contour. Path 3 involves

attraction and deposition of various oxyanions and fluoroanions. Path 4 concerns inclusions of inorganic particles. Path 5 is associated with inclusion of polymers and organic molecules in the emulsion form.

The cathodic current effect kinetics of Rogov and Matthews has been established on the universal electrolyte of KOH and $Na_2SiO_3$. Therefore, the methodology is expected to be applicable in the universal electrolytes with a small amount of foreign additives. When the coating incorporates a high content of additives or an additive causing drastic changes in valve effect, the coating response to cathodic current and its accumulated CIC kinetics ought to be different since the composition and the interfacial rectification change. According to the preceding review on inclusion effects, we may divide the inclusions to three categories. The particles of $SiO_2$, $CeO_2$, $\alpha Al_2O_3$ belong to the category of inert inclusion, since they attenuate the valve effect [93]. The valve effect reduction manifests itself in the earlier current drops in the current–time curves of constant voltage mode when the particle concentration is significant. The coating thickness decreases more rapidly with a higher particle content of the electrolyte [93]. The other extreme is the $Cr_2O_3$ inclusion, which belongs to the category of carrier depletion effect. When incorporated in the coating matrix of *n*-type behavior, the *p*-type $Cr_2O_3$ surface is interfaced with a depleted layer of *p-n* counteracting effect. The details of the depletion effect are not clear at this stage, however, it may cause the voltage runs away during constant current operation [104]. Runaway of *I–V* signifies an extreme consequence of inclusion in obstructing the current passing through a defective oxide coating. The third category involves the additives of polymer particles and organic dye emulsions. We currently have little knowledge of their valve effect influences.

Even though the cathodic current has been identified as a prevailing tool to restrain the discharge damage and control the coating quality, its scope of applicability is still uncertain. Scrutiny over electrolyte compositions of the published PEO studies reveals that inclusions bring a wide range of impacts to the dielectric properties of the coating. Clearly, more research efforts are required to quantify the inclusion influences on *I–V* response of the coating through the concepts of CIC and hysteresis charge.

**Author Contributions:** Conceptualization, D.-S.T.; validation, D.-S.T. and C.-C.C.; resources, D.-S.T. and C.-C.C.; writing—original draft preparation, D.-S.T.; writing—review and editing, C.-C.C.; funding acquisition, D.-S.T. All authors have read and agreed to the published version of the manuscript.

**Funding:** The authors thank Ministry of Science and Technology of Taiwan for financial support through the project MOST 109-2221-E-011-060.

**Data Availability Statement:** Data is contained within the article.

**Conflicts of Interest:** The authors declare no conflict of interest.

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
