# Peer review of "Influences of Growth Species and Inclusions on the Current–Voltage Behavior of Plasma Electrolytic Oxidation: A Review"

_coatings, doi:10.3390/coatings11030270_

Round 1
Reviewer 1 Report
Dear authors, thank You for this interesting review. The language and writing style are sufficient enough.
The paper is well organized and only what is missing is a proper synthetic conclusion section, which would point out the major differences between the approaches and growth paths in a concise way. In the reviewers opinion it would attract more potential readers.
Reviewer 2 Report
Paper : Influences of Growth Species and Inclusions on the Current Voltage Behavior of Plasma Electrolytic Oxidation: A Review, present the state of the art in the field of PEO (electrolytic plasma oxidation (EPO) or microarc oxidation (MAO)) by I-E point of variation.
Please at line L36 explain as first appearence I-V
L36 : mention the reference, is is neccesary
L128, L284, L456 and L499 : insert the reference of the figure, here or in text.
Reviewer 3 Report
In the present manuscript, a lot of material is being reported, but there is relatively little in the way of sifting out the important work and discarding that which contributes little or is contradictory in some way. This is a rather important issue for PEO, which is a field with a high publication rate, but a lot of potential for misunderstanding and confusion. Therefore, I recommend publication of this paper in Coatings after major revision.
Additional comments
[1] The introduction part should be improved. What is the purpose of this review?. As far as I know, there is a comprehensive review published recently in Progress in Materials Science summarizing all aspects of PEO coatings (Progress in Materials Science, 100735, https://doi.org/10.1016/j.pmatsci.2020.100735). The authors should cite that paper and explain what is new in the current manuscript.
[2] As the authors have focused on the effects of electrolyte species on voltage behavior, the effects of organic additives should be discussed. Some electrolyte species led to delay in the appearance of plasma discharges, For example, Effect of sodium benzoate on corrosion behavior of 6061 Al alloy processed by plasma electrolytic oxidation, Surface and Coatings Technology 283, 268-273. So it is suggested to discuss the role of additives in delaying the appearance of plasma sparks. Sodium oxalate, sodium citrate, and EDTA [a-c] were found to change sparking behavior and voltage-time characteristics. So the effects of these additives should be discussed.
(a) Soft plasma electrolysis with complex ions for optimizing electrochemical performance, Scientific reports 7 (1), 1-15.
(b) Hard acid–hard base interactions responsible for densification of alumina layer for superior electrochemical performance, Corrosion Science 170, 108663
(c) Modification of a porous oxide layer formed on an Al–Zn–Mg alloy via plasma electrolytic oxidation and post treatment using oxalate ions, RSC advances 6 (108), 107109-107113
Round 2
Reviewer 3 Report
It can be accepted now.